# Changes in miR-122 and Cholesterol Expression in Chronic Hepatitis C Patients after PegIFN-Alpha/Ribavirin Treatment

**DOI:** 10.3390/pathogens9060514

**Published:** 2020-06-25

**Authors:** Malgorzata Sidorkiewicz, Martyna Grek-Kowalinska, Anna Piekarska

**Affiliations:** 1Department of Medical Biochemistry, Medical University of Lodz, 90-419 Lodz, Poland; martyna.kowalinska@gmail.com; 2Department of Infectious Diseases and Hepatology, Medical University of Lodz, 90-419 Lodz, Poland; anna.piekarska@umed.lodz.pl

**Keywords:** HCV, cholesterol, miR-122, PBMCs, pegIFN-alpha, ribavirin

## Abstract

The hepatitis C virus (HCV) is known as a main etiological cause of chronic hepatitis. HCV infection disturbs cholesterol metabolism of the host, which is frequently observed in patients suffering from chronic hepatitis C (CHC). The course of viral infections remains under strict control of microRNA (miRNA). In the case of HCV, miR-122 exerts a positive effect on HCV replication in vitro. The purpose of this study was to investigate the impact of peginterferon alpha (pegIFN-α) and ribavirin treatments on the expression of miR-122 and the cholesterol level in the peripheral blood mononuclear cells (PBMCs) of CHC patients. We report here that the level of miR-122 expression in the PBMCs decreased after the antiviral treatment in comparison to the pretreated state. Simultaneously, the level of cholesterol in the PBMCs of CHC patients was higher six months following the treatment than it was pretreatment. Consequently, it seems that the decrease of miR-122 expression in the PBMCs of CHC patients is one of the antiviral effects connected with the pegIFN-alpha/ribavirin treatments.

## 1. Introduction

Approximately 180 million people worldwide are known to be infected with the hepatitis C virus (HCV), which is a main etiological cause of chronic hepatitis, cirrhosis, and hepatocellular carcinoma [1]. The development of chronic hepatitis C (CHC) infection is strictly associated with an ineffective immunological response, as well as with lipid disorders of hosts [2]. The HCV particles bind to lipoproteins to form lipoviroparticles (LVPs) [3] and use various host surface receptors, such as the low-density lipoprotein receptor [4] and the scavenger B receptor [5], as important entry factors. HCV RNA replication and virion assembly depend on the products of the cholesterol pathway, such as cholesterol and geranylgeranyl pyrophosphate [6]; HCV assembly requires a platform of cellular lipid droplets that result in a considerable alteration of host lipid metabolism [7]. Therefore, in addition to cirrhosis and hepatocellular carcinoma, HCV infection is also frequently characterized by disorders such as hepatic steatosis and hypocholesterolemia [2].

Although hepatitis C is mainly hepatotropic, the presence of both genomic and antigenomic HCV RNA strands have been documented in the peripheral blood mononuclear cells (PBMCs) of CHC patients [8]. Some studies showed that the hepatitis C virus infects PBMCs and remains a source of virions long after HCV elimination from sera [9,10]. PBMCs proved to be excellent material for observing changes in intracellular cholesterol in CHC patients [11].

miRNAs, i.e., small (19–22 nt) noncoding RNAs, bind mainly to the 3′UTR of pre-mRNAs during post-transcriptional regulation and guide the degradation of target mRNAs. Previous studies showed that miRNAs control the course of viral infections, including the antiviral host response [12]. Previous in vitro studies [13,14] reported that miR-122 exerts a positive effect on HCV replication through a specific interaction with the 5′UTR region of the viral genome. Contrastingly, sequestration of miR-122 in hepatoma cells resulted in a marked loss of replicating HCV RNAs. Being highly abundant in the liver, miR-122 was also described as a factor responsible for the regulation of lipid metabolism in mammalian organisms [15].

The purpose of the current study was to investigate the influence of peginterferon α (pegIFN-α) and ribavirin treatments on serum cholesterol profiles, as well as on the expression of miR-122 and intracellular cholesterol levels in the PBMCs of CHC patients. Our findings indicated that the intracellular cholesterol level in the PBMCs of CHC patients was significantly higher, and miR-122 expression was significantly decreased six months after pegIFN-α/ribavirin treatments than it was before treatment.

## 2. Results

Firstly, serum cholesterol profiles, intracellular cholesterol levels, and miR-122 expressions in the PBMCs were all compared between three independent groups. The groups consisted of 54 untreated CHC patients (NT CHC), 26 CHC patients who were six months post antiviral treatment (T CHC), and 30 healthy donors (HD). The levels of serum total cholesterol (TC), as well as the low-density lipoproteins (LDLs) and high-density lipoproteins (HDLs), were significantly lower in both the untreated and treated CHC patients than in the group of healthy donors (Figure 1A–C). However, these were also found to be slightly elevated in the group of CHC patients who received the antiviral treatment (T CHC) compared those who remained untreated (NT CHC).

Significant differences in intracellular cholesterol (IC) levels and miR-122 expression in PBMC samples were also found between these three independent groups. The NT CHC patients showed lower intracellular cholesterol levels than the HD and T CHC groups (Figure 2A). The miR-122 expressions were lower in the HD and T CHC groups than they were in the NT CHC group (Figure 2B).

The samples taken from the 21 CHC patients before pegIFN-α/ribavirin treatments were then compared with those taken from the same patients six months after treatment cessation. Changes were observed in cholesterol and miR-122 profiles, as well as in HCV RNA levels. As presented in Table 1, the serum HCV RNA before treatment was detected in all 21 CHC patients (mean: 3.25 × 10^5^ IU/mL). The viral load significantly decreased after cessation of therapy and was detectable only in the sera of seven CHC patients (mean: 0.33 × 10^5^ IU/mL).

The strand-specific HCV RNA analysis confirmed the presence of genomic HCV RNA (G) in all PBMC samples collected before treatment. In the same samples, the antigenomic HCV RNA strand (A) was detected in 13 cases (~60%). After treatment, the antigenomic HCV RNA was observed in 4 of 11 PBMC samples (~36%) in which genomic RNA was also detected. The total cholesterol levels and LDL fractions tended to be elevated following treatment (Table 1), and HDL levels tended to be significant higher in CHC patients after treatment compared to baseline. Following treatment, this group of 21 CHC patients demonstrated significantly higher intracellular cholesterol levels and lower miR-122 expressions in their PBMCs.

## 3. Discussion

The impact of the HCV infection on a patient’s lipid disorder is closely related to the HCV life cycle depending on host lipid metabolism [16]. A unique feature of the HCV is that virus cell entry, HCV RNA replication, and virion assembly all depend on the products of host lipid metabolism including cholesterol [7]. Therefore, other than liver steatosis, the main lipid disorder observed in chronic hepatitis C patients is hypocholesterolemia.

It was previously established that the HCV infection may induce the decrease in serum [17,18] and intracellular cholesterol (IC) in PBMCs [11]. The detected hypocholesterolemia and decreased IC levels in the PBMCs of CHC patients in the present study confirmed previous findings.

An insignificantly higher level of the TC, LDL, and HDL fraction was observed in 26 CHC patients following peginterferon α/ribavirin treatments in comparison to 54 untreated CHC patients. The differences were observed more clearly in the group of the same 21 patients who were examined before and after treatment. Similar changes in cholesterol profile following treatment were described by others in studies based on pegIFN-α/ribavirin treatments [19] and those on Daclatasvir/Asunaprevir treatments [20]. It was previously reported that serum cholesterol and the LDL fraction may be treated as a predictive factor of treatment response [21].

A significant increase in intracellular cholesterol (IC) was observed in the PBMCs of CHC patients following treatment, suggesting that the PBMC cholesterol expression levels return to normal following treatment. These are the first data obtained about the post-treatment elevation of intracellular cholesterol in the PBMCs of CHC patients.

The presence of antiviral HCV RNA, a marker of HCV RNA replication, was discovered in PBMCs in previous studies [8,11,22]. It was thus possible to estimate the scale of treatment-induced alterations in the HCV RNA levels by analyzing the sera and the PBMCs of CHC patients before and six months after treatment. A significant reduction or elimination of serum HCV RNA was observed following treatment, along with a reduced expression of genomic and antigenomic HCV RNA in the PBMCs (Table 1). These results confirmed the limited effects of pegIFN-α/ribavirin treatments, as did the presence of genomic and antigenomic HCV RNA strands in PBMCs, especially in patients who had lost HCV RNA from sera; they also suggested the possibility of ongoing HCV replication. Such persistence of HCV was proposed as a possible source of hepatitis reactivation [10,23], and this may be responsible for the limited increase in serum cholesterol observed in our patients.

The interactions between miR-122 and the two binding sites in the 5′-noncoding region of the HCV genome were found to be essential for HCV RNA maintenance and the sequestration of miR-122, leading to a marked loss of HCV RNAs in vitro [13,14]. With this in mind, along with the role of miR-122 in regulation of cholesterol metabolism [15], we also analyzed the effects of peginterferon/ribavirin treatments on miR-122 expressions in the PBMCs of CHC patients. We found that untreated CHC patients presented higher miR-122 expressions in their PBMCs than healthy donors, as previously reported [11]. Similarly, the up-regulation of miR-122 was reported in HIV/HCV coinfected patients in comparison to heathy donors [24].

In the present study, significantly lower levels of miR-122 expression were identified in the PBMCs of treated patients than in untreated patients. The down-regulation of miR-122 was also previously observed in human liver tissue in response to IFN treatments [25]. Similar decreases or normalizations of circulating miR-122 were also reported in patients treated with pegIFN-α/ribavirin [26] or directly-acting antivirals [27]. Since miR-122 was found to facilitate HCV RNA replication [13], it appears that from the analyses observed in our study, the decrease in miR-122 expressions may explain the alteration in the HCV RNA level, especially the reduced detection of the antigenomic HCV RNA strand. The association between the decreased miR-122 expression and the increased cholesterol level proved to be more complicated because earlier studies [15] suggested that miR-122 deletion markedly decreased serum cholesterol. As a result, further studies are necessary to elucidate the exact role of miR-122 in the regulation of cholesterol metabolism.

It seems that both intracellular cholesterol levels and miR-122 expressions in PBMCs can be restored to physiological levels by therapy. The decrease in miR-122 expression in the PBMCs of CHC patients observed following treatment thus appears to be only a single aspect of the antiviral effects connected with pegIFN-α/ribavirin treatments.

## 4. Materials and Methods

This study was approved by the Bioethical Committee of the Medical University of Lodz (RNN/93/07/KB). As a source of sera and peripheral blood mononuclear cells (PBMCs), blood samples were collected from the three groups of people: 54 untreated chronic hepatitis C patients (NT CHC), 26 CHC patients who were six months post pegylated–α–IFN and ribavirin treatment (T CHC), and 30 healthy donors (HD). Written consent was obtained from all subjects in accordance with protocol approved by the Bioethical Committee of the Medical University of Lodz (RNN/93/07/KB).

The PBMCs were isolated by blood centrifugation on a density gradient (Biocoll 1.077, Biochrom) and were used for RNA isolation. Two fractions of RNA: RNA > 200 nt and RNA < 200 nt were extracted from the PBMCs using a mirVanaTM miRNA Isolation Kit (Ambion) according to the manufacturer’s instructions.

The HCV RNA serum was detected by an Amplicor HCV test, version 2.0 (Roche Diagnostics). The genomic and antigenomic strands of HCV RNA in the PBMCs were quantified in RNA fraction > 200 nt using the strand-specific reverse-transcription PCR (RT-PCR) method described by Carreno et al. [8]. Briefly, 40 ng of RNA (>200 nt) was reverse transcribed using 1.25 U of MasterAMP™ Tth DNA Polymerase (Epicentre^®^ Biotechnologies) and 100 nM of antisense primer UTRLC2 (5′–CAAGCACCCTATCAGGCAGT–3′) for genomic strand detection, or sense primer UTRLC1 (5′–CTTCACGCAGAAAGCGTCTA–3′) for antigenomic strand detection. 5 µL of the resulted cDNA was amplified using a Fast Start Universal SYBR Green Master (Roche Diagnostics) and 200 nM of each primer (UTRLC1 and UTRLC2) in the ABIPrism 7900 Sequence Detection System (Applied Biosystems). The relative level of the genomic and antigenomic HCV RNA strands was estimated from Ct values after normalization to Ct values of GAPDH.

The expression of miR-122 was determined in PBMC samples using quantitative real-time reverse transcription PCR (qRT-PCR). Reverse transcription (RT) was performed according to the manufacturer’s instructions on 10 ng of RNA < 200 nt using the TagMan MiRNA Assay specific for miRNA-122 (Applied Biosystems). We then used 3.3 µL of the RT product in the qRT-PCR performed on the ABIPrism 7900 Sequence Detection System (Applied Biosystems) with an miRNA-122-specific primer/probe mix and a TagMan Universal PCR Master Mix (Applied Biosystems) according to the following reaction: 95 °C for 10 min, then 45 cycles 95 °C for 15 s and 60 °C for 1 min. The mean miRNA-122 Ct values were calculated from triplicate reactions and normalized to mean snRNA U6 Ct values. The expression of miR-122 relative to snRNA U6 was calculated using the formula 2^-ΔΔCt^, according to the Applied Biosystems guidelines.

To determine the relative intracellular cholesterol (IC) level, the intracellular cholesterol level in the PBMCs was evaluated using the cholesterol assay kit Cholesterol Chod-PAP (BIOLABO S.A., France) per the manufacturer’s recommendations; the results were normalized to the protein concentration in lysates. Serum total cholesterol (TC), high-density lipoprotein-cholesterol (HDL-C), low-density lipoprotein-cholesterol (LDL-C) levels were measured enzymatically using an Olympus AU 640.

Statistical analyses were performed with STATISTICA 8.0 PL software (Statsoft). The results that originated from the three independent groups (HD, NT CHC, and T CHC) were compared using the Kruskal–Wallis test. To compare the results obtained before and after treatment from the same group of 21 CHC patients, the Mann–Whitney U test was used. The *p*-values < 0.05 were considered statistically significant. The data in the figures and table are presented as mean values ± standard deviation (SD).

## 5. Conclusions

Our findings demonstrated that pegIFN-α/ribavirin therapies restore the PBMC intracellular cholesterol levels and miR-122 expressions in CHC patients to physiological levels. It is possible that the post-treatment decrease in miR-122 expressions observed in the PBMCs of CHC patients is one part of the range of antiviral effects demonstrated by anti-HCV treatment.

## Figures and Tables

**Figure 1 pathogens-09-00514-f001:**
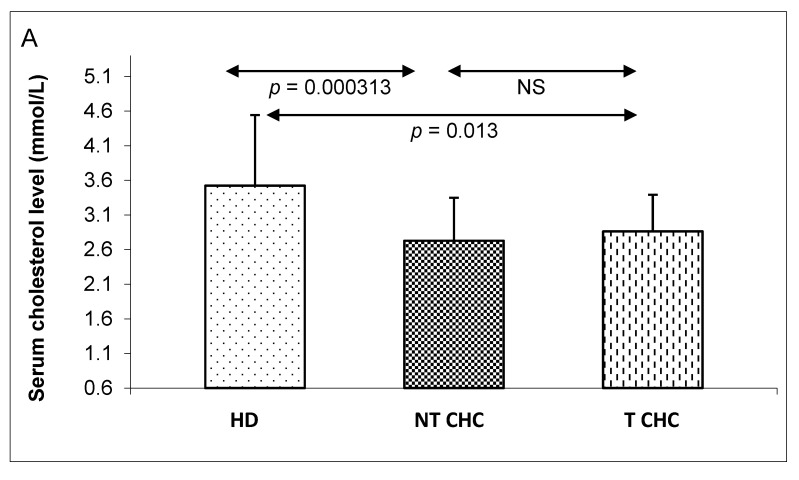
A comparison of serum total cholesterol (**A**), low-density lipoprotein (LDL) (**B**) and high-density lipoprotein (HDL) (**C**) levels between three independent groups: healthy donors (HD), untreated CHC patients (NT CHC), and CHC patients six months after pegIFN-α/ribavirin treatments (T CHC) (via the Kruskal–Wallis test). NS = non-significant.

**Figure 2 pathogens-09-00514-f002:**
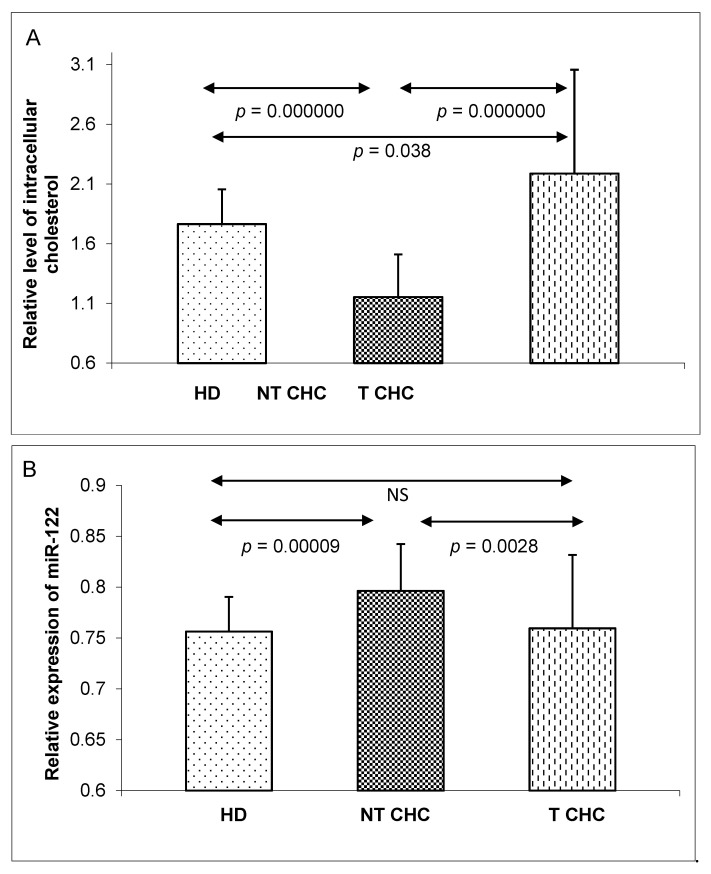
A comparison of intracellular cholesterol levels (**A**) and miR-122 expressions in PBMCs (**B**) between three groups: healthy donors (HD), untreated CHC patients (NT CHC), and CHC patients six months after pegIFN-α/ribavirin treatments (T CHC) (via the Kruskal–Wallis test). NS = non-significant.

**Table 1 pathogens-09-00514-t001:** A comparison of selected parameters in the samples taken from the same 21 CHC patients before (CHC Before) and after (CHC After) the pegIFN-α/ribavirin treatments.

Analyzed Parameters	CHC Before	CHC After	*p* Value +
Serum HCV RNA positive (%)	21 (100)	7 (29.2)	ND
Serum HCV RNA load (× 10^5^ IU/mL)	3.245 ± 2.99	0.3333 ± 0.48	0.0008
ALT (IU/L)	67.8 ± 50.5	35.32 ± 18.5	0.003
G/A HCV RNA strand (PBMCs)	21/13	11/4	ND
TC (mmol/L)	2.669 ± 0.72	2.8476 ± 0.57	0.1098
LDL (mmol/L)	1.4629 ± 0.5	1.5938 ± 0.5	0.1023
HDL (mmol/L)	0.8062 ± 0.27	0.9162 ± 0.23	0.0137
IC (PBMCs)	1.8986 ± 0.56	2.4047 ± 0.82	0.0228
miR-122 (PBMCs)	0.8027 ± 0.06	0.7070 ± 0.17	0.0005

+ = Mann–Whitney U test; ND = not determined; ALT (IU/L) = serum alanine aminotransferase concentration; G/A HCV RNA strand = number of genomic-strand-positive to antigenomic-strand-positive PBMCs samples; serum concentration of TC = total cholesterol; LDL = low-density lipoproteins; HDL = high-density lipoproteins; IC = relative intracellular cholesterol level in PBMCs; miR-122 = relative miR-122 expression in PBMCs.

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
