# Peer review of "Changes in miR-122 and Cholesterol Expression in Chronic Hepatitis C Patients after PegIFN-Alpha/Ribavirin Treatment"

_pathogens, 2020, doi:10.3390/pathogens9060514_

Round 1

Reviewer 1 Report

  1. Authors should discuss the mechanism of miR122 on cholesterol metabolism.
  2. Authors should describe the methods of the measurement of miR122 in detail.
  3. Authors should uniform microRNA or miRNA throughout the text.

Author Response

  1. The association of decreased miR-122 expression with the increase level of cholesterol observed in our study is not easy to explain. The study of Esau et al showed opposite situation: miR-122 deletion markedly decreased serum cholesterol. Probably, to elucidate the exact role of miR-122 in regulation of cholesterol metabolism further study is necessary. This issue is now discussed in lines 140-145.
  2. The method of the miR-122 measurement is now described in details between lines 172 and 181.
  3. The abbreviation miRNA is introduced in line 8 and consistently used to end the manuscript.

Reviewer 2 Report

The authors should be rewritten this manuscript in a format in Pathogens MDPI.

e.g. numbering wrong, all Figures and table should be changed. statistical data, p value should be given in your data.

Please see the author guidelines and improve this manuscript. I want to see the revised manuscript.

Author Response

The manuscript has been rewritten in the format of Pathogens MDPI including figures, table, statistical data and citations.

Reviewer 3 Report

This paper evaluates the activity of chronic HCV infection on the metabolism of cholesterol. In the manuscript, the profile of serum cholesterol and the amount of cholesterol in peripheral blood mononuclear cells has been detailed in patients with chronic HCV. 

The authors found that the serum levels of total cholesterol, LDL cholesterol, HDL cholesterol were greater in patients with chronic HCV who underwent combined antiviral therapy (pegIFN plus ribavirin) in comparison with those who did not receive antiviral therapy. 

Another finding of the manuscript was the intracellular amount of cholesterol (in peripheral blood mononuclear cells, PBMCs) was greater in patients with chronic HCV who received antiviral therapy in comparison with those who were untreated. The relative expression of miR-122 was greater in those patients who were untreated compared with those who received antiviral therapy. 

I suggest to shorten the Section Discussion. I suggest to give more details in table 1. It remains unclear (not only to the busy reader) some evidence reported in the Table. As an example, the meaning of TG, relative IC, relative miR-122. I suggest to include more details in the legend of the Table. 

It remains unclear the links between the various experiments. The connection between the evidence reported in Figure 1, and Figure 2a and 2b is not clear. The authors should give an explanation on this in the Section Discussion. The pieces of evidence reported in the Section results of the manuscript do not appear to be included with a logical approach.  

Author Response

  1. It was possible to slightly shorten the Discussion by transferring part of the information on relation between HCV infection and lipid metabolism to the Introduction (now lines 23-30) and by removing the part connected with studies of Akuta et al.
  2. More details is given in the legend of Table 1 (lines 84-88)
  3. In section Results we changed the description of experimental groups to better explain the connection between experiments and data presented in Fig 1 (lines 51-57) and Fig 2 (lines 65-68) and in Table 1 (lines 75-80 and 90-97).

Round 2

Reviewer 2 Report

The authors addressed the queries raised and improved the manuscript for the publication.

Reviewer 3 Report

The manuscript looks now nicer; the authors have appropriately addressed in the text my comments.